# Effective Prediction of Concrete Constitutive Models for Reinforced Concrete Shear Walls under Cyclic Loading

**DOI:** 10.3390/ma17081877

**Published:** 2024-04-18

**Authors:** Quoc Bao To, Jiuk Shin, Sung Jig Kim, Hye-Won Kim, Kihak Lee

**Affiliations:** 1Deep Learning Architecture Research Center, Department of Architectural Engineering, Sejong University, 209 Neungdong-ro, Gwangjin-gu, Seoul 05006, Republic of Korea; 2Department of Architectural Engineering, Gyeongsang National University, Jinju 52828, Republic of Korea; 3Department of Architectural Engineering, Keimyung University, 1095 Dalgubeol-daero, Dalseo-gu, Daegu 42601, Republic of Korea; sjkim4@kmu.ac.kr; 4Senior Research Officer, Earthquake Hazards Reduction Center, National Disaster Management Research Institute, Ulsan 44412, Republic of Korea

**Keywords:** RC shear walls, concrete constitutive models, cyclic loading, FE model

## Abstract

One of the most challenging elements of modeling the behaviour of reinforced concrete (RC) walls is combining realistic material models that can capture the observable behaviour of the physical system. Experiments with realistic loading rates and pressures reveal that steel and concrete display complicated nonlinear behaviour that is challenging to represent in a single constitutive model. To investigate the response of a reinforced concrete structure subjected to dynamic loads, this paper’s study is based on many different material models to assess the advantages and disadvantages of the models on 2D and 3D RC walls using the LS-DYNA program. The models consisted of the KCC model and the CDP model, which represented plasticity and distinct tensile/compressive damage models, and the Winfrith model, which represented plasticity and the smeared crack model. Subsequently, the models’ performances were assessed by comparing them to experimental data from reinforced concrete structures, in order to validate the accuracy of the overall behaviour prediction. The Winfrith model demonstrated satisfactory results in predicting the behaviour of 2D and 3D walls, including maximum strength, stiffness deterioration, and energy dissipation. The method accurately predicted the maximum strength of the Winfrith concrete model for the 2D wall with an error of 9.24% and for the 3D wall with errors of 3.28% in the X direction and 5.02% in the Y direction. The Winfrith model demonstrated higher precision in predicting dissipation energy for the 3D wall in both the X and Y directions, with errors of 6.84% and 6.62%, correspondingly. Additional parametric analyses were carried out to investigate structural behaviour, taking into account variables such as concrete strength, strain rate, mesh size, and the influence of the element type.

## 1. Introduction

### 1.1. Background

The material known as concrete is characterized by its brittleness, and the rate at which it is subjected to loading has a substantial role in determining its resistance to external forces and its propensity for damage, particularly in terms of softening behavior [1]. Extensive study has been undertaken over the course of several decades to investigate the modeling of concrete behavior when subjected to static or quasi-static loading [2]. The persistent problem in the numerical simulation of concrete structures is in the creation of efficient and accurate constitutive models. In addition, there is a need for models that accurately depict the mechanical behavior of materials under the influence of external forces.

Reinforced concrete (RC) constructions are frequently employed in earthquake-prone regions. The lateral force-resisting mechanism in these projects consists of either moment frames, structural walls, or a combination of both [3,4,5,6]. To guarantee that the structural system can undergo sufficient inelastic deformation for satisfactory performance, modern design standards, like the current ACI 318 specification (ACI 318: building code requirements for structural concrete), establish specific criteria for reinforced concrete (RC) elements in areas susceptible to seismic activity [7]. Concrete shear walls are commonly acknowledged as prominent load-bearing structures. Utilising concrete shear walls as lateral load-resisting constructions has numerous benefits. These factors encompass the ability to achieve a high level of cost efficiency and outstanding performance when subjected to horizontal forces, such as those caused by earthquakes. Concrete shear walls are frequently used to withstand lateral loads because of their exceptional capacity to endure horizontal forces. When designing seismic-resistant structures, engineers strive to meet three important design criteria: ductility, strength, and stiffness. The efficacy of concrete shear walls in fulfilling these criteria has been well documented [8,9,10,11].

The Finite Element Method (FEM) is a well utilised numerical technique employed in many disciplines to solve differential equations or for mathematical modelling purposes. LS-DYNA is a FEM analysis programme [12]. While LS-DYNA offers a range of material models to predict concrete behaviour, it is difficult to undertake comprehensive concrete experiments to accurately establish parameters. In order to alleviate the challenge of addressing this matter, LS-DYNA offers uncomplicated input concrete models for simulating the performance of concrete based on fundamental strength test data [13,14]. Several studies examining the behaviour of material models in predicting the response of concrete buildings to different loading rates have been documented in the literature [15,16,17,18]. The concrete material model takes into account the effect on the elasticity of concrete before it cracks, as well as the successive plastic states when cracking begins [19,20].

Prior studies on the simulation of reinforced concrete shear wall structures under monotonic and cyclic loads have employed two-dimensional finite element (2D-FE) models [21,22]. The response of reinforced concrete (RC) structures to impact and seismic loads has been investigated through the utilisation of three-dimensional finite element (3D-FE) models [23,24,25,26]. Various types of concrete structures, such as columns, frames, beam-to-column connections, slabs, bridge girders, bridge decks, and moment-resisting frame systems, have been studied extensively [27]. The previous investigations have explored the use of reinforced or fiber-reinforced concrete, as well as post-tensioning techniques. The research has been conducted by several authors [28,29,30]. 

### 1.2. Research Purpose

Prior studies have extensively assessed the effectiveness of concrete constitutive models for RC shear walls with simplified shapes, namely barbell and rectangular types, which are commonly known as 2D wall types in this paper. However, there is a limited amount of research available on how concrete constitutive models affect the lateral resistance capabilities of reinforced concrete shear walls with flanged and irregular section types, also referred to as 3D walls. This encompasses both planar and nonplanar walls, including their distinctions in relation to uniaxial and biaxial stress, torsional properties, and other relevant factors. Hence, the primary objective of this study was to examine the behavior of reinforced concrete walls using different concrete material models by means of comparing them to experimental findings. The aim of this study was to evaluate the efficacy of various concrete constitutive models in accurately reproducing essential laboratory material characterization data. This technique focused on emphasising the important differences between the models, while also identifying the necessary observable behaviours that the models should be able to reproduce. Consequently, the concrete models were utilised to assess the influence of the constitutive models on the estimated response. This evaluation was conducted in several scenarios, such as different concrete strengths, strain rates, mesh sizes, and element types.

## 2. Constitutive Modeling of Concrete

Porous and brittle characteristics are important features and are the most complicated behaviours of concrete when subjected to different loading situations [31]. LS-DYNA contains several material models that simulate the mechanical behaviour of concrete. The three concrete constitutive models are used herein as follows: the KCC model (MAT072R3), the CDP model (MAT273), and the Winfrith model (MAT085).

### 2.1. The KCC Model (MAT072R3)

The MAT072R3 is a concrete material model commonly referred to as the Karagozian and Case Concrete model (KCC) [32,33,34,35,36]. This model is classified as an isotropic material model that incorporates plasticity and damage. It was developed by Malvar et al. [37]. The remaining parameters can be automatically determined by adjusting the default values of a typical concrete with a compression strength of 45.6 MPa. The uniaxial compressive strength is the only parameter used in the model, as it is adequate for the calibration process [38].

The failure boundaries of concrete are defined by three surfaces in a three-dimensional principal-stress space: the maximum failure surface, yield surface, and residual failure surface. These surfaces are illustrated in Figure 1. The graphic displays the maximum failure surface, yield surface, and residual failure surface in the principal-stress space. This space is separated into several intervals, with $f_{c}$ representing the concrete compressive strength.

The KCC model utilises simple functions to define three separate failure surfaces, each characterised by a distinct set of parameters. These surfaces are the maximum shear failure surface (described in Equation (Equation 1)), the residual failure surface (described in Equation (Equation 2)), and the yield failure surface (described in Equation (Equation 3)). The parameters for each surface are denoted as ($a_{0}$, $a_{1}$, $a_{2}$) for the maximum shear failure surface, ($a_{0f}$, $a_{1f}$, $a_{2f}$) for the residual failure surface, and ($a_{0y}$, $a_{1y}$, $a_{2y}$) for the yield failure surface [39].
(1)$$\Delta \sigma_{m}=a_{0}\left(\frac{p}{a_{1}\left(a_{2}\times p\right)}\right)$$
(2)$$\Delta \sigma_{m}=a_{0f}\left(\frac{p}{a_{1f}\left(a_{2f}\times p\right)}\right)$$
(3)$$\Delta \sigma_{m}=a_{0y}\left(\frac{p}{a_{1y}\left(a_{2y}\times p\right)}\right)$$
where $\Delta \sigma_{m}$ and *p* are the difference in the principal stresses and the pressure in an element, respectively.

### 2.2. The CDP Model (MAT273)

The Concrete Damage Plasticity (CDP) model [40,41] describes the failure mechanism of concrete under applied loads. The CDPM model, developed by Grassl [40], incorporates both plasticity and damage-based models, which can be either isotropic or anisotropic. There are two methods to specify tensile strain softening and determine the appropriate crack width, which can be utilised either by the user or by default values. The post-peak compressive performance is described by an exponential function with a softening control parameter, which has a default value of $\left(\varepsilon_{fc}=10^{-4}\right)$. This is shown in Figure 2a. The occurrence of brittle damage is more likely when a smaller value of $\left(\varepsilon_{fc}\right)$ is employed. The strain rate effect is taken into account when STRFLG = 1 and is not considered otherwise (STRFLG = 0).

The post-peak tensile behaviour in this model might exhibit linearity, bilinearity, or exponentiality. The researchers utilised the bilinear damage model depicted in Figure 2b for this study due to its accurate evaluation of the experimental data [41]. The authors created a constitutive model that combines damage mechanics and plasticity to analyse the failure of concrete structures. The objective was to acquire a model that accurately depicts the significant attributes of the failure process of concrete under multiaxial loading. In the given diagram, $G_{f}$ represents the fracture energy, which is determined by the area under the strain softening curve. On the other hand, $w_{f}$ denotes the maximum tensile inelastic strain, and its value may be approximated using the formula $w_{f}=4.444G_{f}/f_{t}$. The model response was compared to five groups of experiments reported in the literature. For each group of experiments, the physical constants Young’s modulus *E*, Poisson’s ratio ν, tensile strength $f_{t}$, compressive strength $f_{c}$, and tensile fracture energy $G_{t}$ were adjusted to obtain a fit for the different types of concrete used in the experiments as mentioned in the source [41].

### 2.3. The Winfrith Model (MAT085)

The Winfrith model, known as MAT_WINFRITH_CONCRETE in LS-DYNA, utilises a smeared-crack model that was first established by Broadhouse and Neilson [42] for modelling purposes. The Winfrith model necessitates the user to specify the unconfined compression and tensile strength as essential input parameters. The model consists of 32 input values and provides the ability to generate specific model input parameters based on the concrete compressive strength, which serves as the sole input. The information provided is derived from a four-parameter failure surface established by Ottosen [43]. The primary advantage of the Winfrith model is its ability to accurately determine the precise location and dimensions of a crack using a binary output database. The crack width can be determined by analysing the region beneath the curve of uniaxial tensile stress versus crack width, which is equal to the fracture energy ($G_{f}=G_{f0}/10$), $G_{f0}$ is the base of fracture energy. It depends on the size of the structure member. This equation is cited in Equations (2.1–7) from the CEB-FIP Model Code [44] as illustrated in Figure 3. The Winfrith input parameter, RATE, specifies whether the investigation of the strain rate effect is conducted (RATE = 0) or not (RATE = 1, 2). The distinction between RATE = 1 and 2 lies in the fact that RATE = 2 investigates enhanced crack algorithms. The unconfined tensile strength is determined by Equation (Equation 4). The estimation can be derived from the uniaxial compressive strength. The elastic modulus of typical concrete is calculated using Equation (Equation 5).

This model assumes a compressive behaviour that is both elastic and perfectly plastic. The yield surface of the model is constructed using the four-parameter plastic surface described in Equations (Equation 6) and (Equation 7) by Ottosen et al. [43].
(4)$$f_{t}=1.4{\left(\frac{f_{c}}{10}\right)}^{2/3}$$
(5)$$E_{c}=4700\sqrt{f_{c}}$$
(6)$$F_{i}\left(I_{1},J_{2},\cos3\theta \right)=a\frac{J_{2}}{{\left(f_{c}\right)}^{2}}+\lambda \frac{J_{2}}{f_{c}}+b\frac{I_{1}}{f_{c}}-1$$
(7)$$\lambda =\left\{\begin{array}{c} k_{1}\cos\left[\frac{1}{3}\cos^{-1}\left(k_{2}\cos3\theta \right)\right]\left(\cos3\theta ⩾0\right)\\ k_{1}\cos\left[\frac{\pi }{3}-\frac{1}{3}\cos^{-1}\left(-k_{2}\cos3\theta \right)\right]\left(\cos3\theta <0\right) \end{array}\right.$$
where θ is the Lode angle (Lode parameter), which is a function of the stress deviator’s third invariant and is used to discriminate between distinct shear stress states in three dimensions (3-D); *a*, *b*, $k_{1}$, $k_{2}$ are parameters which are a function of $f_{t}/f_{c}$, which can be auto-generated in LS-DYNA; the variables $f_{t}$ and $f_{c}$ are the tensile strength and compressive strength of concrete, respectively; and $E_{c}$ (MPa) is Young’s modulus.

## 3. Analytical Methodology Validation

### 3.1. Experimental Investigation

Pakiding et al. [45] conducted experiments on 2D walls, while Beyer et al. [46] conducted experiments on 3D walls. The primary objective of the investigations was to ascertain the capacity of a 2D wall to endure significant deformations prior to failure, as well as to examine the bending characteristics of a 3D wall under bi-directional loading in various directions. Typically, 2D and 3D walls are used as lateral bracing elements in tall buildings to enhance their load-bearing capacity. These empirical discoveries lay the groundwork for future investigations into the seismic response of walls.

The dimensions of the 2D wall are 6730 mm in height, 1830 mm in length, and 250 mm in thickness. The dimensions of the foundation block for the wall were 1520 mm in length, 7620 mm in width, and 610 mm in height. The wall consisted of eight longitudinal bars measuring 22 mm in diameter and two longitudinal bars measuring 9.5 mm in diameter. The dimensions of the 3D wall were 3490 mm in height, 1600 mm in length, and 150 mm in thickness. The wall was reinforced with continuous bars extending from the foundation to the collar, ensuring longitudinal reinforcement without any lap splices. The total area of the vertical reinforcement was approximately $A_{s}=3281$ mm^2^, as depicted in Figure 4.

The concrete’s compressive strength was 43 MPa for the 2D wall and 45 MPa for the 3D wall, as determined by the material properties. The ASTM A615 and ASTM A706 standards were employed to reinforce the steel with a nominal yield strength of 414 MPa in the 2D wall. In addition, the 3D wall included reinforcing bars that complied with Eurocode 8 standards for “Class C” grade steel, with yield strengths of 488 MPa for 12-mm diameter bars and 518 MPa for 6-mm diameter bars.

### 3.2. Constitutive Models of Materials

The concrete constitutive model’s material parameters were calibrated in a consistent manner. The uniaxial compressive strength, $f_{c}$, was calculated from material testing conducted in conjunction with the experiments of the structural component. Poisson’ s ratio, ν, was set equal to 0.2 for all analytical models. The element removal accounts for the material failure of concrete were related to cover spalling.

The steel material used in this investigation was represented by the MAT#03 model in LS-DYNA, specifically the *MAT_ PLASTIC_KINEMATIC model. This model is suitable for simulating both isotropic and kinematic hardening plasticity. Figure 5 illustrates the elastic–plastic characteristics of the *MAT_PLASTIC_KINEMATIC model. In this figure, $l_{0}$ represents the original length of the uniaxial tension specimen, *l* represents the length after deformation, and $E_{t}$ represents the slope of the bilinear stress–strain curve. A cyclic loading approach was used with a hardness parameter of 0.3, as stated in the reference [47]. The Poisson’s ratio was uniformly set to 0.3 for all the analyses, as it was also a requirement for the continuum-based beam elements. The material properties for concrete and steel reinforcements are provided in Table 1.

The *MAT_ELASTIC material model was employed to simulate the behaviour of the base and top components in LS-DYNA. For the sake of simplicity and to focus on the behaviour of the walls, the loading section at the top and the material used for the wall footing were assumed to be elastic. The analyses utilised Rayleigh damping, with a pre-established damping ratio of 1% for the walls. Under these conditions, the cyclic loads were applied at a sufficiently low rate (strain rate ranging from $10^{-3}$ s^−1^ to $10^{-1}$ s^−1^) to confirm that the RC walls could be subjected to earthquake loading.

### 3.3. Contact and Boundary Conditions

The *CONSTRAINED_LARGRANGE_ IN_ SOLID option was utilised to incorporate the reinforcements into the concrete walls. The nodes located on the bottom face of the base were subjected to fixed supported boundary constraints. This study employed the Dynamic Implicit analysis technique, as outlined in the publication of [48]. The global response was conceptualised as a dynamic problem, where the equations of motion over time were solved using a central difference method. The equations of motion were solved by utilising the updated geometry of the mesh at every stage of the investigation.

## 4. Experimentation-Based Numerical Analysis and Comparison

### 4.1. Evaluation of 2D Walls under Cyclic Loading

#### 4.1.1. Backbone Curves and Hysteresis

This section focuses on the validation of several concrete material models. The efficacy of the material models in accurately predicting the behaviour of the 2D wall has been proven in several studies [49,50,51,52,53]. The research findings are compared to empirical data. The analysis was performed on the 2D wall specimens that were tested by Pakiding et al. [45]. The wall’s configuration details are illustrated in Figure 6a. It features two prestressing tendon bundles, each composed of five strands, which were utilised to apply a prestressing force of 1561 kN. The wall incorporated sixteen #1 bars, each with a diameter of 22 mm, in addition to vertical #3 bars with a diameter of 10 mm in the web and boundary regions of the section. The wall also featured horizontal reinforcing #4 bars with a diameter of 10 mm. An examination was conducted on the wall sample using a predetermined cyclic horizontal displacement pattern. This pattern was applied at a specific height of 3810 mm from the foundation block until the point of damage occurred.

The FE models of the 2D wall are depicted in Figure 6b–d. An eight-node solid element (ELFORM = 1) was used to model the concrete wall, including its top part and foundation. The RC walls were reinforced utilising a beam element with cross-section integration, namely the Hughes–Liu beam model. This formulation method is renowned for its efficiency, precision, and effectiveness when dealing with significant deformations. The concrete wall was partitioned into two sections using mesh sizes of 25.4 mm and 50.8 mm, respectively. The beams were also meshed with a size of 25.4 mm. This meshing configuration was employed in sensitivity analyses, as reported by Shin et al. [54] in their study on retrofitting. The foundation and upper components were modelled using a larger mesh size (e.g., 100 mm) compared to the primary structural elements (i.e., wall, beam) as shown in Figure 6c.

The analysis utilised three specific models to represent concrete material in a 2D wall that was exposed to cyclic loads. The experimental results were compared to the backbone and hysteresis curves derived from the FE model in order to confirm the accuracy of the suggested FEM, as shown in Figure 7. The KCC concrete model accurately predicted the peak load and initial stiffness with 8.68% and 4.82%, respectively, at a drift ratio of 3%. However, no pinching was observed in this model. The CDP concrete model accurately predicted the pinching effect, as demonstrated in these figures. However, it failed to accurately capture the strength deterioration when compared to the experimental results with 21.44% and 9.67% for initial stiffness and peak load, respectively. The Winfrith concrete model, however, provided an acceptable estimation of the maximum load and the constriction effect when compared with experimental results with 9.24% of maximum strength as shown in Table 2.

The energy dissipation capacity of a structure is a crucial measure that has a considerable impact on energy-based seismic design since it reflects the structural performance. The value of this index is greatly influenced by the structural components that comprise the entire system. In order to optimise analysis time, each cyclic load was performed only once per cycle for the analytical models. Due to the challenge of directly comparing the energy dissipation with the tests, the total dissipation energy was calculated by summing the area of the envelope of the hysteresis curve, as demonstrated in Equation (Equation 8) [55]. The graph is depicted in Figure 8. The CDP model closely approximated the experimental values for dissipation energy, with a deviation of only 0.06%. The KCC and Winfrith models exhibited a discrepancy in their predictions of 41.69% and 28.08%, respectively. The key characteristics of the hysteresis diagrams are shown in Table 2, including the initial stiffness, maximum strength, and energy dissipation.
(8)$$E=\sum_{n=1}^{\infty }P\times \Delta_{n}$$
where *P* is the load in the envelope curve and $\Delta_{n}$ is the micro-displacement in the envelope curve.

#### 4.1.2. Behaviour of Failure

The concrete damage mechanism and damage pattern were investigated using the maximum principal strain fringe, which can be evaluated in LS-DYNA. The areas of highest flexural stresses in reinforcement bars can be determined by analysing the maximum principal strain contours of models. Contour ranges are provided for each example due to the variation in strain ranges across different material models. Figure 9 illustrates the extent of damage to the two-dimensional wall as determined using numerical simulation. This image presents the assessment of concrete and reinforcement damage by utilising a colour palette that corresponds to the maximum principal strain fringe. The purpose is to accurately identify the extent of damage in the wall. The KCC model’s concrete and rebar are located at the heart of the main colour palette in this picture. Therefore, it appears that the KCC model is incapable of detecting the specific failure of concrete or the highest tensile area in rebars, as depicted in Figure 9a. This aligns with the model’s trend of stress continuously rising, as illustrated in Figure 7a. In contrast, the CDP model displays a distinct colour area for concrete at the highest point of the colour scale in the maximum principal strain fringe. Consequently, the diagram clearly illustrates that the failure takes place in the vicinity of the corner of the wall. Due to the limited colour range of rebars, the depiction of places with the highest flexural stresses in beams is not entirely precise. The CDP model is capable of identifying crushing in certain components at the base. This observation aligns with the weakening of strength shown in its force-displacement diagram. The source of this weakening can be attributed to the damage formulas utilised in the model, as shown in Figure 9b. However, in the Winfrith model, it is possible to reasonably estimate the location of failure and the regions of greatest tensile stress in rebars. This is achieved by detecting the colour area for concrete and rebars at the highest point of the colour palette in the maximum principal strain fringe. The damage manifests in the vicinity of the base by altering the shape of the contours, as depicted in Figure 9c.

### 4.2. Evaluation of 3D Walls under Cyclic Loading

#### 4.2.1. Backbone Curves and Hysteresis

The 3D reinforced concrete or U-shaped walls are frequently employed as lateral load-bearing elements in RC structures due to their ability to offer strength and stiffness in all horizontal directions, as well as their suitability for accommodating lift shafts or staircases. The subsequent validation investigation pertains to a three-dimensional reinforced concrete wall specimen, specifically referred to as specimen TUA test by Beyer et al. [46].

The configuration specifics of the wall are illustrated in Figure 10a. The wall’s height and thickness were 2650 mm and 150 mm, respectively, as depicted in the figure. The reference is to Figure 10b. This model incorporates the concept of bidirectional loading, which refers to the application of force in both the X and Y directions. The wall incorporated a total of twenty-two #6 bars, each with a diameter of 12 mm, in the border areas. Additionally, there were twenty-eight #7 bars, each with a diameter of 6 mm, in the web regions of the section. In addition, the transverse reinforcement #7 had a diameter of 6 mm and was spaced 50 mm apart along the margins of the wall and in the web sections. This reinforcement was comprised of ties with a diameter of 6 mm and a spacing of 125 mm. An examination was conducted on the wall sample using a predetermined cyclic bidirectional displacement pattern. This pattern was applied at a height of 3350 mm and 2950 mm above the base for loading in the X and Y directions, respectively, until damage occurred. The FE model of the wall specimen is illustrated in Figures. Refer to Figure 10c,d. A solid element with eight nodes (ELFORM = 1) was employed to simulate the concrete wall, top part, and base. Furthermore, the calculation of the RC wall reinforcements was performed utilising a beam element that incorporates cross-section integration, specifically the Hughes–Liu beam method. For finite element analysis, it is crucial to determine the correct mesh size and element type. In this case, the concrete wall was divided into two parts with mesh sizes of 25.4 mm and 50.8 mm, while the beams were similarly meshed with a size of 25.4 mm. In order to reduce computational time, the base and top sections were simulated using a higher mesh size (e.g., 50 mm and 100 mm) compared to the principal structural elements (i.e., the wall and the beam).

A 3D wall analysis was performed on three concrete models subjected to cyclic loading. The analytical results for the 3D wall, obtained using different concrete models (KCC, CDP, and Winfrith models), are compared to the corresponding experimental data in Figure 11. The KCC model did not adequately anticipate the peak strength and pinching effect in both directions. The KCC model had errors of 25.58% and 14.54% for initial stiffness and peak strength in the X direction, respectively, as well as errors of 8.8% and 14.61% for initial stiffness and peak load, respectively, in the Y direction. The CDP concrete model accurately predicted the peak strength and pinching effect in the Y direction with errors of 8.6%, 3.81% for initial stiffness and peak load, respectively, as well as the positive region in the X direction with errors of 3.04% and 2.31% for initial stiffness and peak load, respectively. However, it performed poorly in capturing the negative peak strength in the X direction as shown in Figure 11b. The method effectively predicted the maximum strength of the Winfrith concrete model with an error of 3.28% for the X direction and 5.02% for the Y direction as shown in Table 3 and Table 4, while it greatly overstated the pinching effect in both directions.

Given the extensive utilisation of many components in models that absorb much of the energy from earthquakes, it was crucial to evaluate the model’s capacity to disperse energy. Figure 12 illustrates a contrast between the total dissipation energy obtained from numerical simulation and experimental testing conducted by Beyer et al. [46] in their study on quasi. The Winfrith model had superior accuracy in predicting dissipation energy in both the X and Y directions, with deviations of 6.84% and 6.62%, respectively. However, the CDP and KCC models exhibited better predictions in the Y direction with errors of 4.53% and 7.23%, respectively, compared to the X direction with errors of 22.67% and 42.44%, respectively. The essential characteristics of the hysteresis diagrams are presented in Table 3 and Table 4, which include important factors such as the initial stiffness, peak strength, and energy dissipation for the 3D wall in the X and Y directions, respectively.

#### 4.2.2. Behaviour of Failure

The analysis of this model will focus on the damage mechanism and damage pattern, which will be assessed using the maximum principal strain fringe. This study employed a colour palette based on the maximum principal strain fringe to determine the extent of damage in the wall, encompassing both the damaged concrete and rebars. The colour palette used in this study aligns with the established standard for assessing damage in 2D walls. Figure 13 illustrates the extent of the damage to the 3D reinforced concrete wall as determined using numerical simulation. The KCC model identified the specific failure area and predicted the regions with the highest tensile stress. However, its accuracy was limited since the colour range observed for concrete and rebars was only concentrated at the highest point of the primary colour spectrum, as shown in Figure 13a. However, the CDP model accurately identified the failure location of concrete and the areas of greatest tensile stress in rebars. This was achieved by recognising the specific colour range for concrete and rebars at the highest point of the colour palette in the maximum main strain fringe. The CDP model determined that concrete crushing occurs when elements are removed from the corner of the wall, as shown in Figure 13b. On the other hand, the Winfrith model accurately predicted the location of failure and the areas of maximum tensile stress in the rebars by comparing the colour areas of concrete and rebars and the colour palette of maximum principal strain fringes, as depicted in Figure 13c. The strength degradation is consistent with the damage formulas, as depicted in Figure 11.

## 5. Parametric Studies

Given the FE model’s ability to accurately predict the response of RC walls, including their peak strength and failure behaviour, additional parametric tests were undertaken on the structural performance of the RC walls using the validated FE model. An investigation was conducted to examine the impact of concrete strength on three distinct models: the KCC, CDP, and Winfrith models. The Winfrith concrete model was selected to study parameters such as strain rate, mesh size, and element type due to its superior predictive capability for model behaviour. This experiment was carried out using both 2D and 3D walls.

### 5.1. Effectiveness of Concrete Compressive Strength ($f_{c}$)

The study examined the impact of concrete compressive strength by utilising pushover analysis as a means to expedite the analysis process. The relationship between the compressive strength of concrete and the load-displacement curves was demonstrated in Figure 14 and Figure 15. The provided examples were examined by gradually increasing the intensity from 25% to 100%, which corresponds to a range of $f_{c}$ values from 43.4 MPa to 86.8 MPa for 2D walls and from 45 MPa to 90 MPa for 3D walls. Overall, raising the value of $f_{c}$ improved the lateral load of the specimen in all situations. Based on the 2D wall plots, increasing $f_{c}$ by 100% (from 43.4 to 86.8 MPa) resulted in a 16.3%, 19.1%, and 18.9% improvement in lateral strength for the KCC, CDP, and Winfrith models, respectively, at a lateral drift of 3.5%. The CDP and Winfrith models exhibited similar improvements in lateral load as strength increased. Specifically, the lateral load increased by 50%, equal to 10.1% and 10.4% for the CDP and Winfrith models, respectively. Similarly, a 100% increase in strength corresponded to a 19.1% and 18.9% increase in lateral load for the CDP and Winfrith models, respectively, at a lateral drift of 3.5%. The comprehensive data are presented in Table 5. Figure 15 shows the effect of compressive strength in 3D walls. When the compressive strength increased 25%, the lateral strength improved by 3.1%, 3.5%, and 3.3% in the X direction as well as 3.2%, 3.0%, and 3.4% in the Y direction for the KCC, CDP and Winfrith models, respectively. In addition, the lateral strength had significant improvements by 8.0% and 7.8% in the X direction and Y direction, respectively, for three concrete models when the compressive strength increased 50%. Moreover, the lateral strength increased by 11.4%, 13.6%, and 12.8% in the X direction at a lateral drift of 1.5%, and by 10.6%, 19.7%, and 21.8% in the Y direction at a lateral drift of 2.5% for the KCC, CDP, and Winfrith models, respectively, when the compressive strength ($f_{c}$) was increased by 100%. The CDP and Winfrith models provided an estimation of the enhancement in lateral load when the strength was increased from 25% to 100% in the X direction at a lateral drift of 1.5% and in the Y direction at a lateral drift of 2.5%. The comprehensive data are presented in Table 6 and Table 7.

### 5.2. Strain Rate Effects

The Winfrith concrete model considers the presence of a smearing crack. This break forms when the maximum main stress exceeds the maximum tensile stress, and it is oriented perpendicular to the direction of the stress. Despite the initial formation of a fracture, the model remained capable of withstanding shear, compressive, and tensile forces. Thus, the Winfrith model employs a smeared fracture to better accurately replicate the behaviour of concrete under tensile stress. An experiment was carried out to examine the strain rate impacts of Winfrith models at different rates, specifically RATE = 0, RATE = 1, and RATE = 2. Figure 16 and Figure 17 display the force-displacement outcomes of two-dimensional and three-dimensional walls. The hysteretic loops observed in the 2D wall did not accurately anticipate the pinching and peak strength because they did not account for the rate effects. Although RATE = 2 successfully caught the pinching effect, it was unable to accurately anticipate the maximum strength due to its enhanced crack algorithm. In contrast, when RATE = 1 was used, it provided an acceptable prediction for both the pinching and peak strength. When it comes to the 3D wall, a prediction of the pinching effect can be reasonably made with RATE values of 0, 1, and 2. However, while both RATE values of 1 and 2 were able to accurately represent the peak strength, RATE value of 0 could not, especially in the X direction where the peak strength was higher. When analysing the Y direction, the model saw that RATE = 0, RATE = 1, and RATE = 2 all exhibited a pinching effect similar to that observed in the X direction. However, only RATE = 1 accurately predicted the peak strength.

### 5.3. Mesh-Size Effects

The mesh size is a critical factor in FEA and directly affects the accuracy and quantity of meshes required for element meshing. The mesh size not only impacts accuracy but also directly influences the computational time of the model. Consequently, studying the impact of the mesh size is of utmost significance. By selecting an appropriate mesh size, the model may optimise computational efficiency and provide precise outcomes. The initial models utilised a mesh size of 25 mm. The influence of the mesh size on the cyclic behaviour of components was assessed in both 2D and 3D walls. The study examined the impact of different mesh sizes on the 2D and 3D walls. Two mesh sizes, 25 mm and 50 mm, were used for the 2D wall, while three mesh sizes, 12.5 mm, 25 mm, and 50 mm, were used for the 3D wall. The results are shown in Figures. The range of figures from Figure 18 and Figure 19. The experimental results were compared to the hysteresis and backbone curves derived from the FEA in order to confirm the impact of mesh size on the outcomes, as depicted in Figures. The range of figures are from Figure 20 and Figure 21. The force-displacement equations remained consistent regardless of the variation in mesh size for both the 2D and 3D walls. This demonstrates that the Winfrith model is not influenced by changes in mesh size of the concrete elements, as the strength and stiffness of the RC components are not affected by such alterations. Although the Winfrith model did not undergo substantial changes based on mesh size, the authors advise that the mesh size should be smaller than 50 mm.

### 5.4. Concrete Element Effects

The specific behaviour was replicated by utilising an eight-node reduced integrated solid element. LS-DYNA utilises four frequently employed solid element types, including the fully integrated S/R solid element. This element is specifically designed for cases with low aspect ratios and employs precise and efficient formulations. The element types are denoted by the values −2 and −1 for the ELFORM parameter. The typically utilised solid element types in LS-DYNA are the fully integrated S/R solid element with ELFORM = 2 and the constant stress solid element with ELFORM = 1. The examinations of these four sorts of elements are contrasted with the experimental data from a 3D wall study by Beyer et al. [46] in Figure 22. Upon comparison, it is evident that nearly all of the models successfully predicted the resistance of the specimen. However, the constant stress solid element yielded the most precise estimation of the test specimen’s hysteretic behaviour. In addition, Figure 23 illustrates that among the four models with varying mesh sizes, the constant stress solid element provided the least amount of time required for calculations. The simulation time for a fully integrated S/R solid element was at least 1.15 times longer than the simulation time for a constant stress solid element. This part demonstrates a high level of precision and effectiveness in terms of calculation.

## 6. Conclusions

A selection of cyclic tests was chosen to analyse the behaviour of multiple concrete constitutive models. Three LS-DYNA concrete models, namely the KCC model (MAT072R3), the CDP model (MAT273), and the Winfrith model (MAT085), were selected to assess their effectiveness in predicting the seismic behaviour of reinforced concrete structures. Tables were generated to condense the key findings from each research, including starting stiffness, peak strength, and energy dissipation. The subsequent enumeration encapsulates several significant findings:1.Among the three material models examined in this study, the KCC model accurately projected the cyclic performance of both the flexural and shear components, including the deterioration of strength and stiffness for the 2D wall. The KCC model exhibited lower efficacy compared to the CDP and Winfrith models in predicting the degradation of peak strength and stiffness in the 3D wall. Both models, the Winfrith model and the examined components, showed reasonable results in predicting the peak strength, stiffness deterioration, and energy dissipation. In summary, the Winfrith model proved to be the most dependable for analysing the behaviour of shear walls. Therefore, it is recommended to limit the use of the KCC and CDP models when simulating the reaction of shear walls.2.The failure mode of the examined components, which was accurately predicted by the CDP and Winfrith models, involved the crushing of concrete. The places experiencing the highest tensile stress were identified using fringes indicating maximum tensile strain. The destruction of concrete was demonstrated by removing elements once they met the erosion criterion. The maximum principal strain contours of the KCC, CDP, and Winfrith models can identify the areas with the highest flexural stresses in beams.3.The CDP and Winfrith models were similar regarding the improvement of the force-resistance of 2D and 3D walls when varying the concrete strength. The strain rate had a significant effect on seismic response in which the RATE = 1 was appropriate for RC walls. Even though the mesh size and element type had a negligible effect on peak strength and pinching effects, they significantly affected the time analysis of the model. Hence, the optimal choice for the numerical simulation was a mesh size of 25 mm and an element type of ELFORM = 1.4.In light of the findings from this study, the authors intend to utilise the Winfrith and CDP concrete models in future investigations to examine the behaviour of several shapes of reinforced concrete (RC) walls, including L, T, and I shapes. These particular designs were chosen due to their widespread application in contemporary building practices. However, this research has a limitation in that it ignores the friction ratio in the bonding between rebars and concrete. Instead, it replaces it with a contact function in the LS-DYNA software throughout the simulation process. This will be improved and taken into account in further investigations.

## Figures and Tables

**Figure 1 materials-17-01877-f001:**
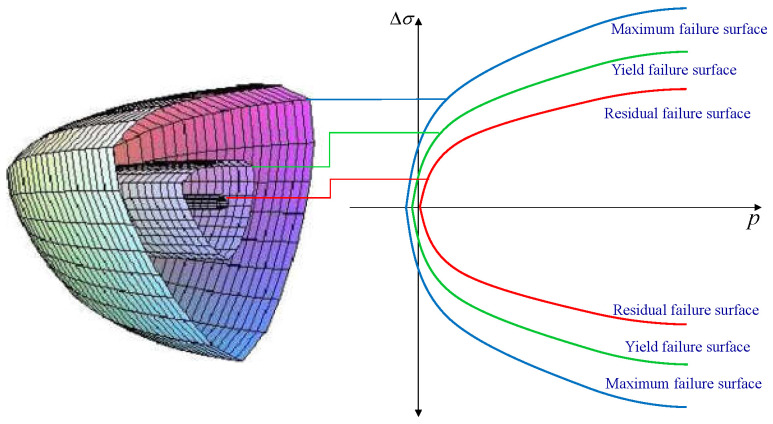
Boundary surfaces in the space of major stresses.

**Figure 2 materials-17-01877-f002:**
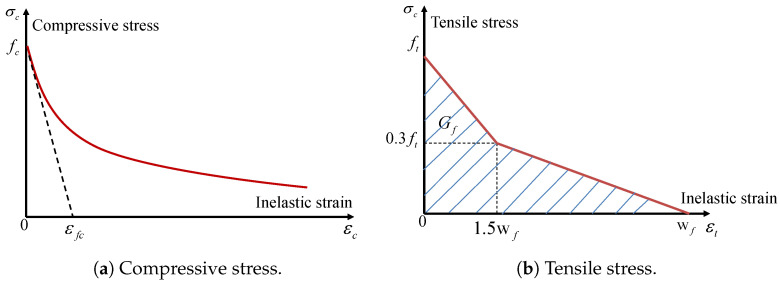
The strain-softening behaviour of the CDP model.

**Figure 3 materials-17-01877-f003:**
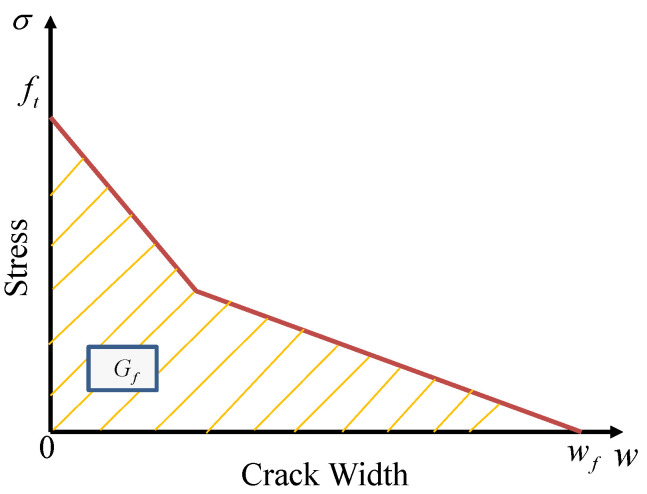
Response of crack strain softening.

**Figure 4 materials-17-01877-f004:**
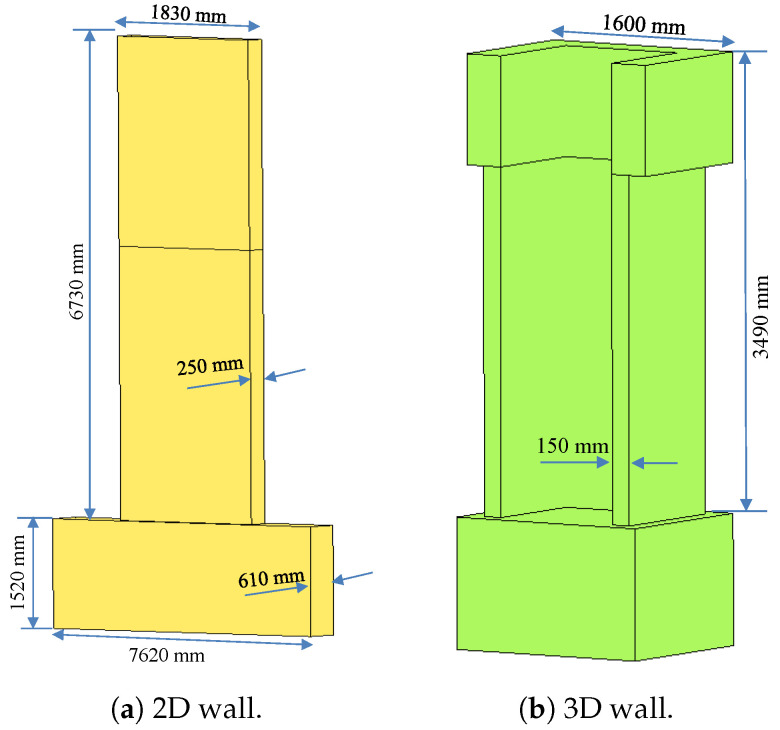
Dimension for test of 2D and 3D walls.

**Figure 5 materials-17-01877-f005:**
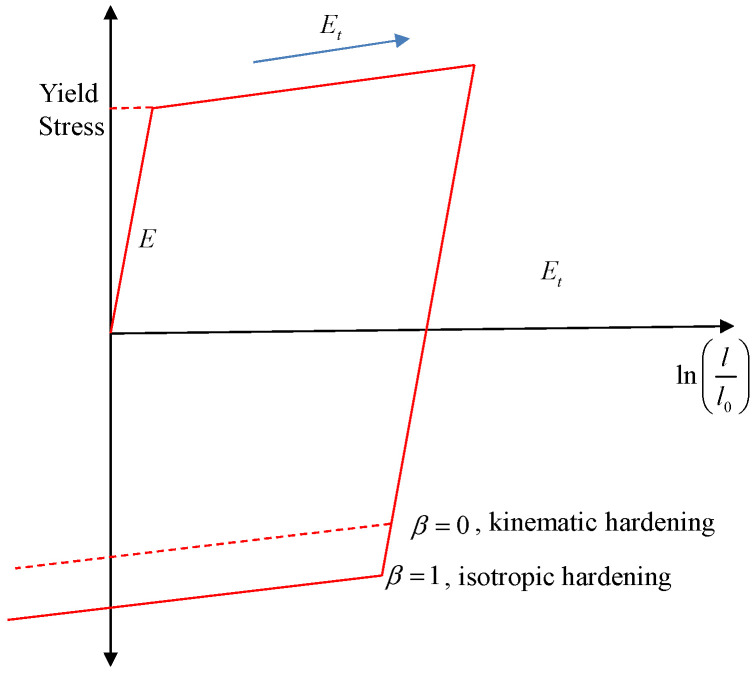
Elastic–plastic behaviour with isotropic and kinematic hardening.

**Figure 6 materials-17-01877-f006:**
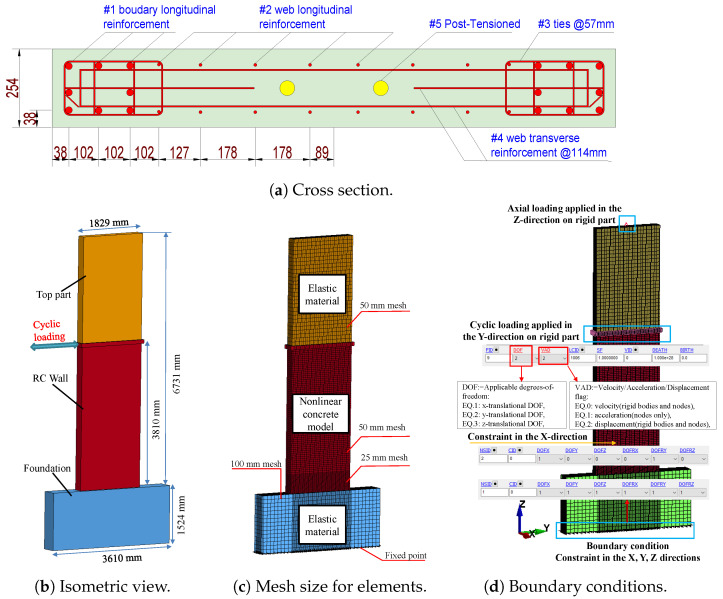
Geometric and FE model for the 2D wall.

**Figure 7 materials-17-01877-f007:**
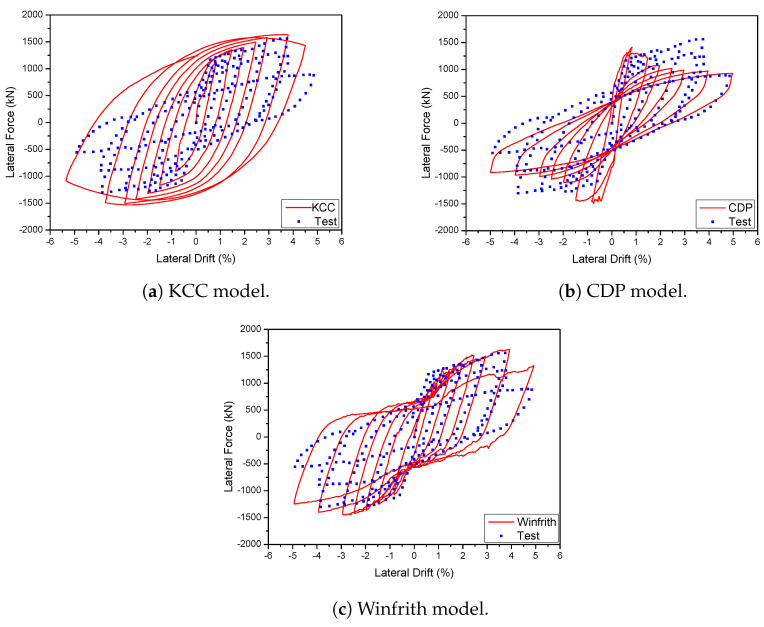
Load-displacement curve prediction for the 2D wall.

**Figure 8 materials-17-01877-f008:**
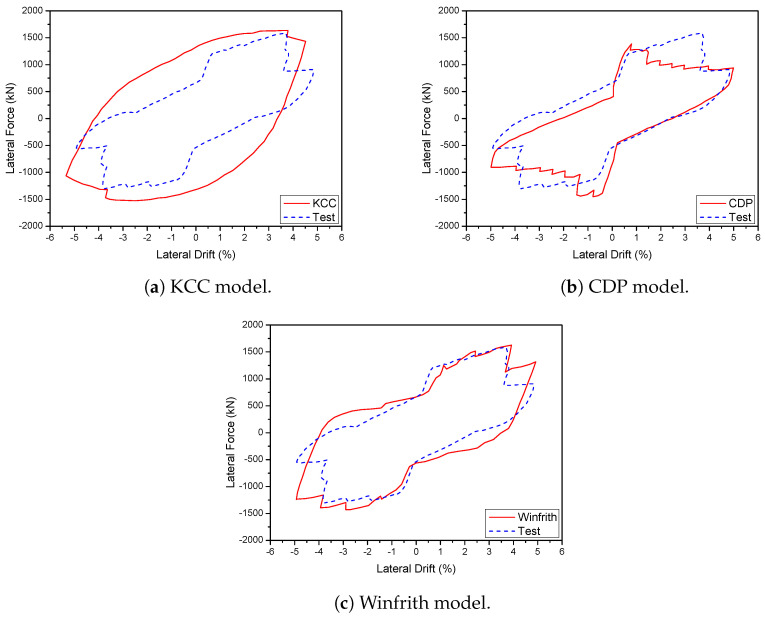
Energy dissipation envelope curve predicted for the 2D wall.

**Figure 9 materials-17-01877-f009:**
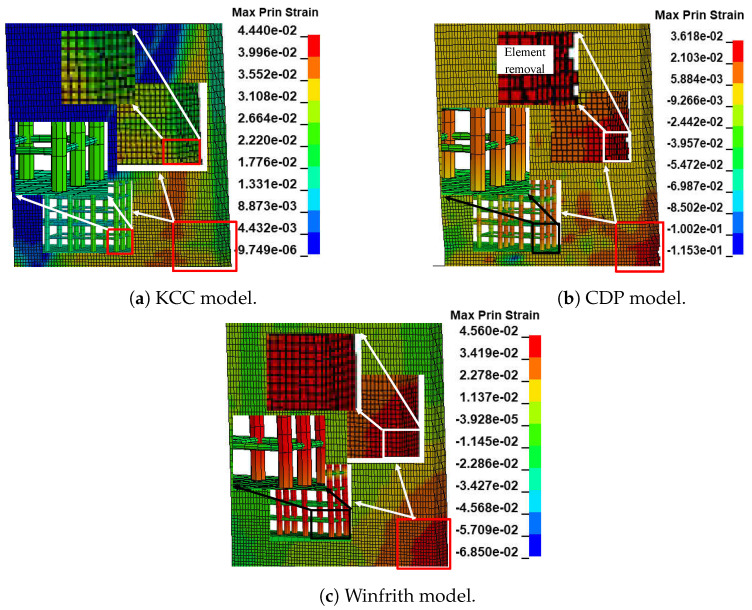
The FE model demonstrates the structural damage to the 2D wall.

**Figure 10 materials-17-01877-f010:**
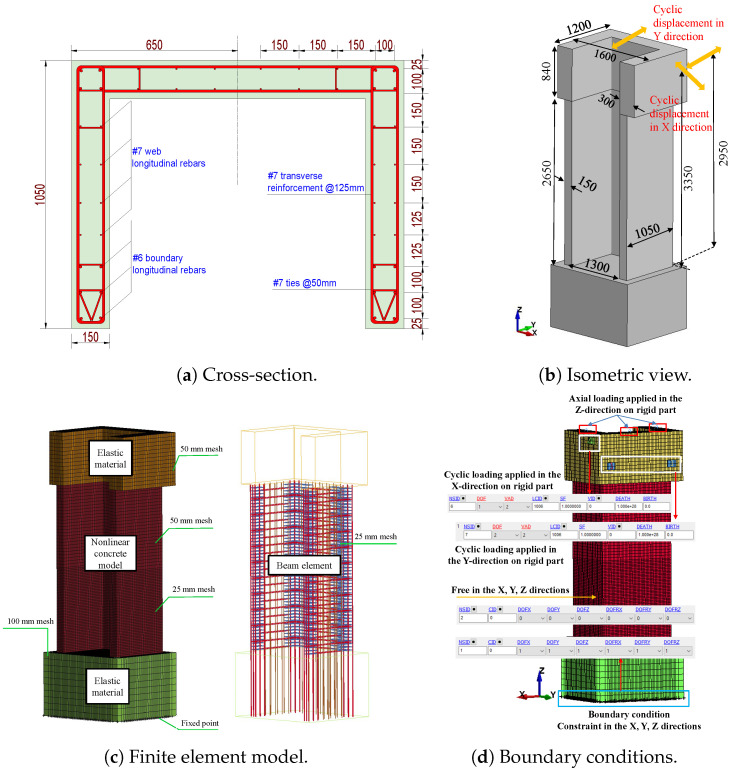
Geometric reinforced details and FE model for the 3D wall.

**Figure 11 materials-17-01877-f011:**
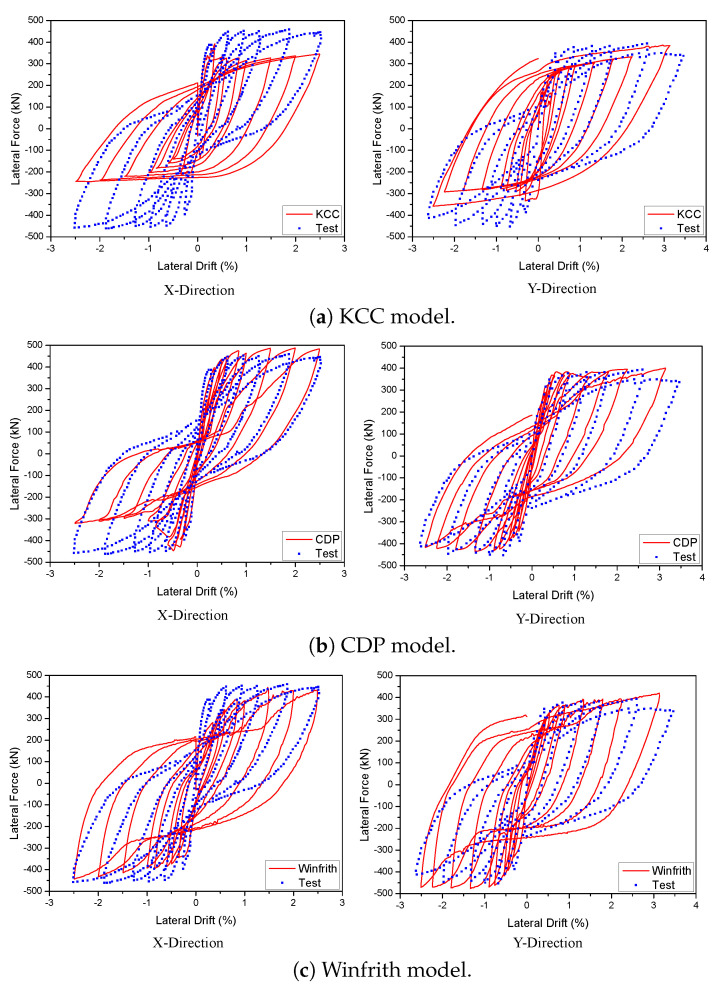
Load-displacement curve prediction for the 3D wall.

**Figure 12 materials-17-01877-f012:**
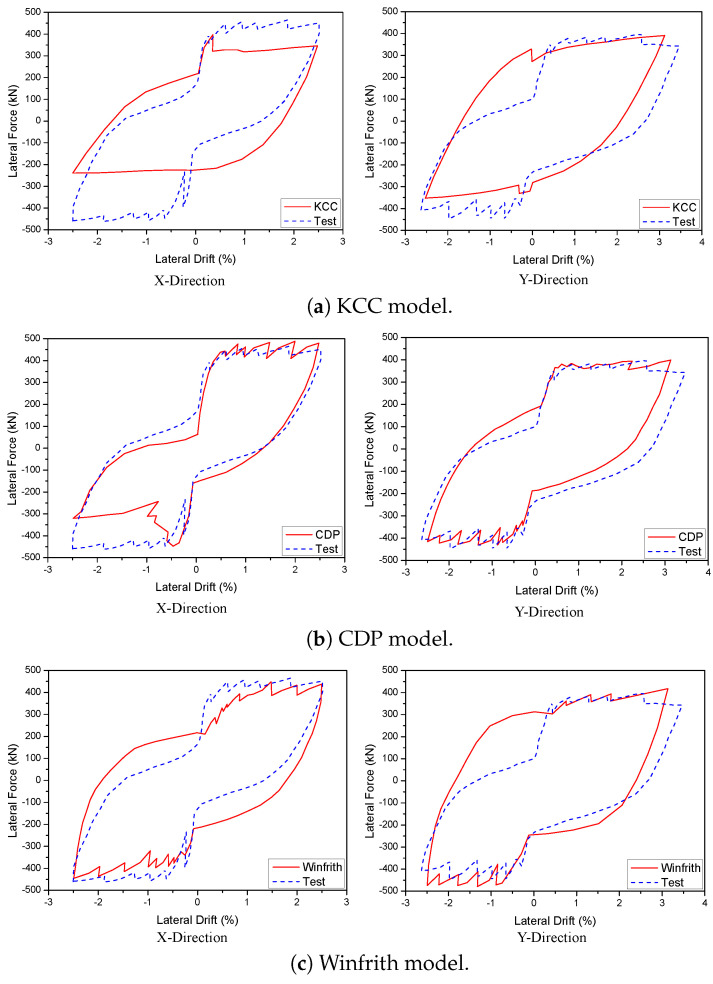
Energy dissipation envelope curve predicted for the 3D wall.

**Figure 13 materials-17-01877-f013:**
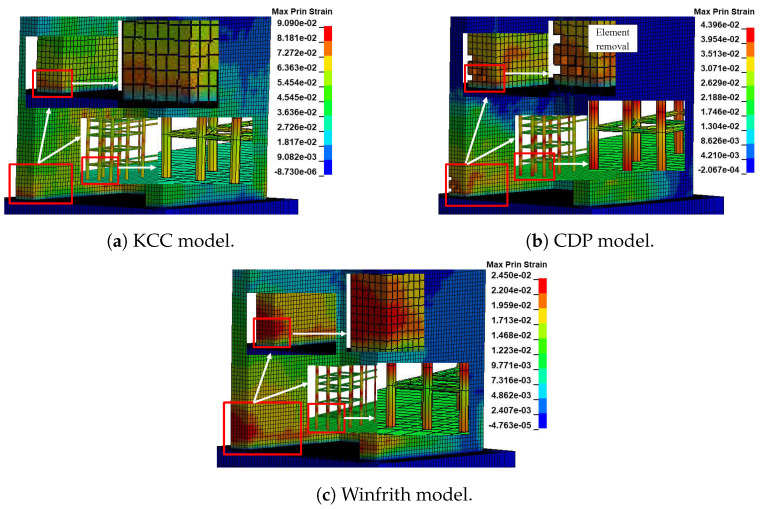
The FE model demonstrates the structural damage to the 3D wall.

**Figure 14 materials-17-01877-f014:**
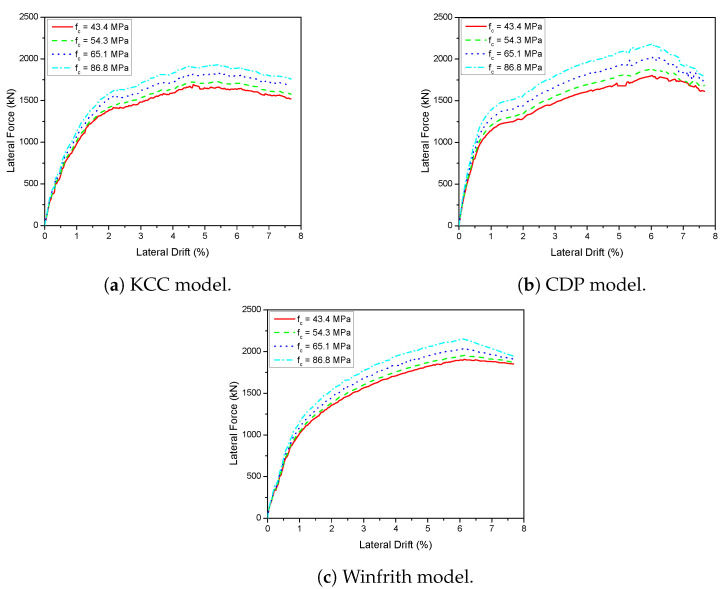
The influence of the compressive strength of concrete on the load-displacement behaviour of the 2D wall.

**Figure 15 materials-17-01877-f015:**
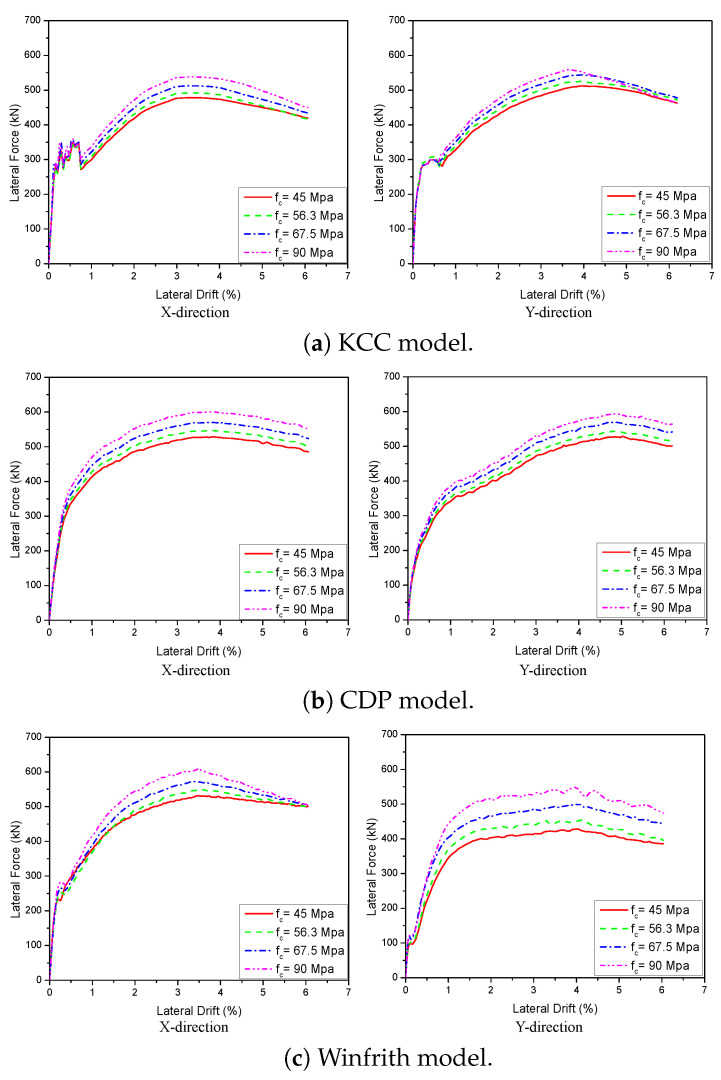
The influence of the compressive strength of concrete on the load-displacement behaviour of the 3D wall.

**Figure 16 materials-17-01877-f016:**
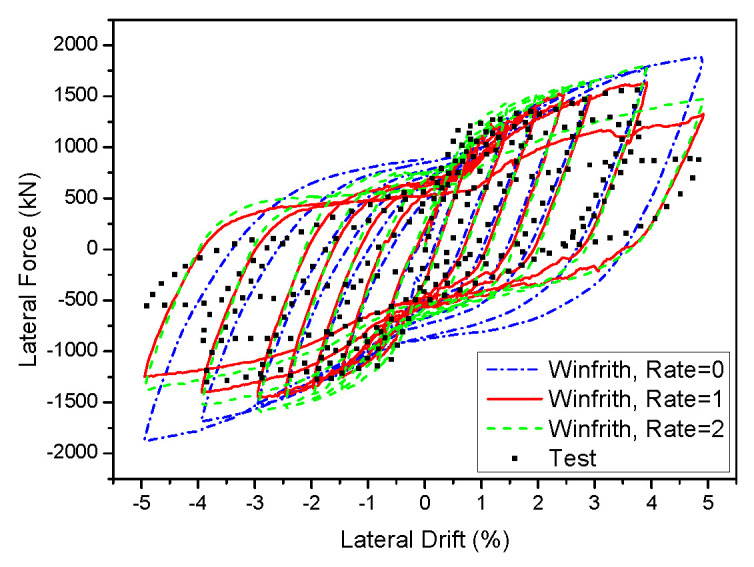
Influence of various strain rate on the 2D wall in the Winfrith model.

**Figure 17 materials-17-01877-f017:**
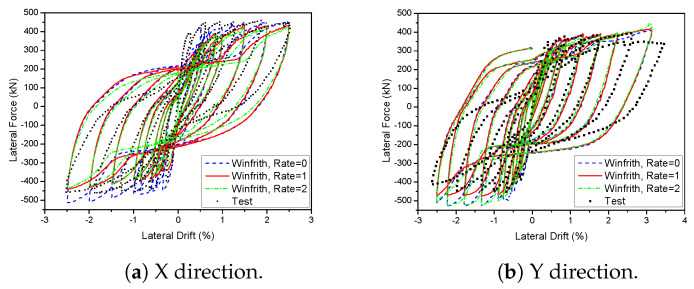
Influence of various strain rate on the 3D wall in the Winfrith model.

**Figure 18 materials-17-01877-f018:**
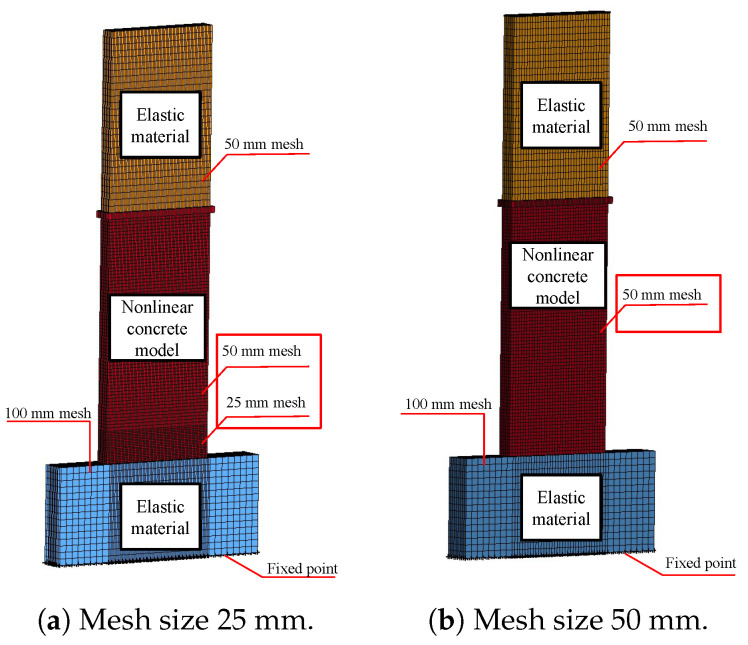
Finite element model of mesh size effect for the 2D wall.

**Figure 19 materials-17-01877-f019:**
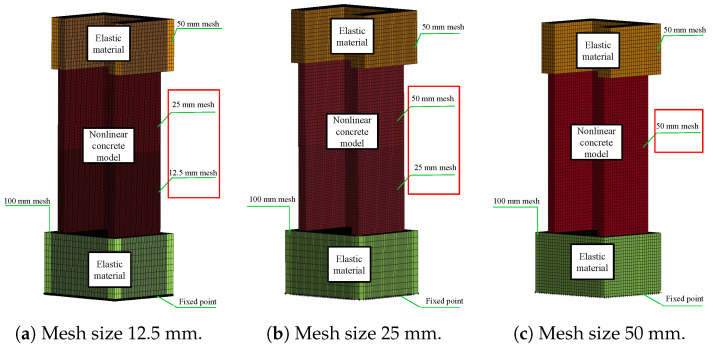
Finite element model of mesh size effect for the 3D wall.

**Figure 20 materials-17-01877-f020:**
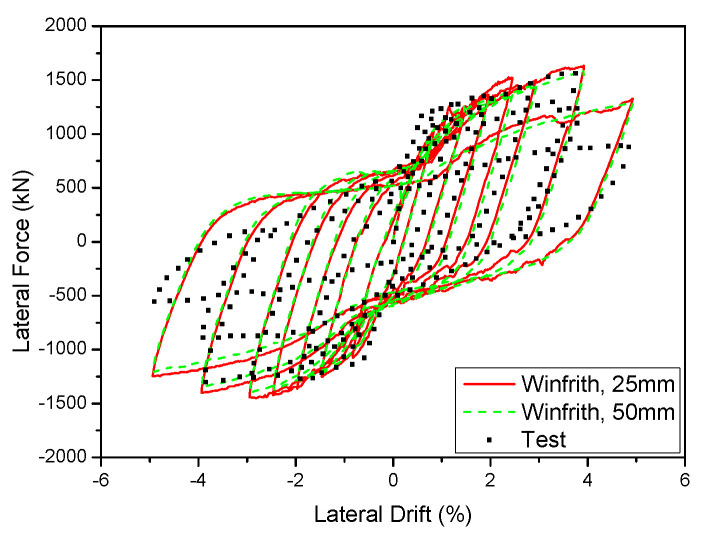
Influence of various mesh sizes on the 2D wall in the Winfrith model.

**Figure 21 materials-17-01877-f021:**
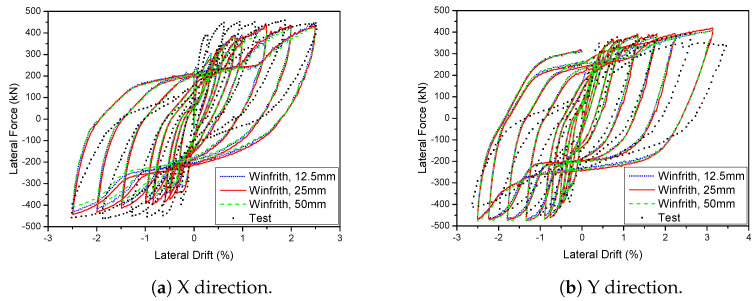
Influence of various mesh sizes on the 3D wall in the Winfrith model.

**Figure 22 materials-17-01877-f022:**
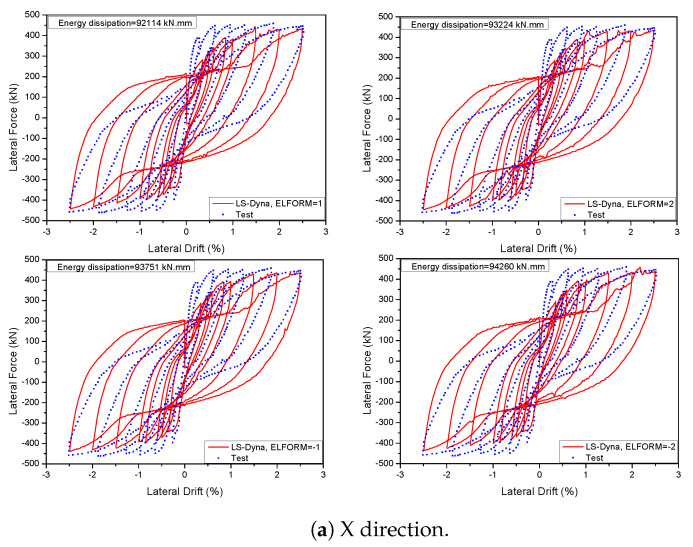
Influence of different element types on the 3D wall in the Winfrith model.

**Figure 23 materials-17-01877-f023:**
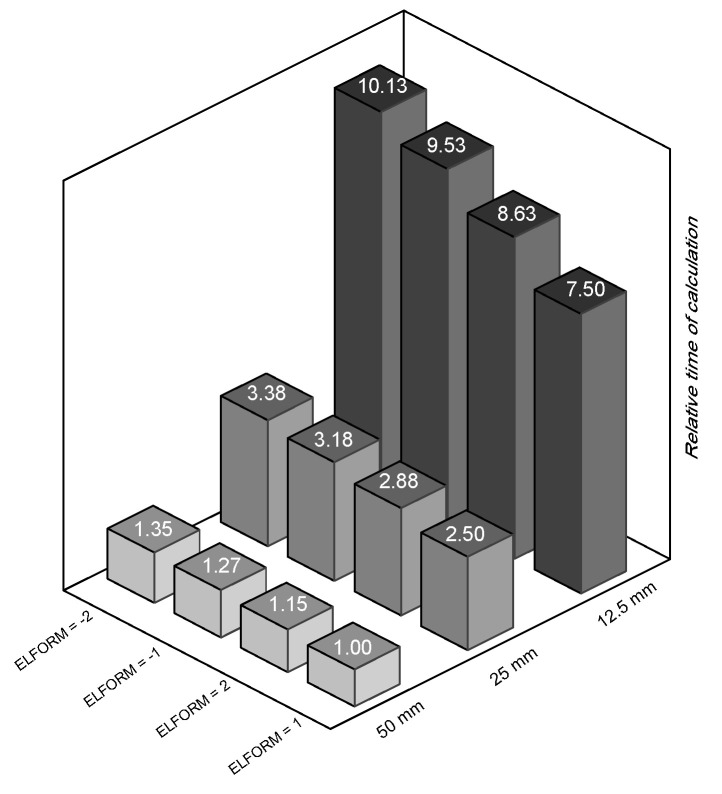
The relationship between calculation time and element types and mesh sizes.

**Table 1 materials-17-01877-t001:** The mechanical characteristics of steel reinforcements and concrete.

Specimen	Concrete		Steel Reinforcements					
	$f_{c}$ (MPa)	ρ (g/mm^3^)	Bar Type	$E_{s}$ (MPa)	ρ (g/mm^3^)	ν	$f_{y}$ (MPa)	$f_{u}$ (MPa)
Pakiding et al. [45] 2D Wall	43.4	0.0023	#1	200,000	0.00783	0.3	519	744
			#2	200,000	0.00783	0.3	473	742
			#3	200,000	0.00783	0.3	473	742
			#4	200,000	0.00783	0.3	441	683
			#5	200,000	0.00783	0.3	1675	2038
Beyer et al. [46] 3D Wall	45	0.0023	#6	200,000	0.00783	0.3	519	744
			#7	200,000	0.00783	0.3	518	681

It is important to mention the following: $f_{c}$ represents the compressive strength of the concrete, ρ represents the mass density, ν represents Poisson’s ratio, $E_{s}$ represents Young’s modulus, and $f_{y}$ and $f_{u}$ represent the yield stress and ultimate strength of the rebar, respectively.

**Table 2 materials-17-01877-t002:** Hysteresis diagram parameters pertaining to the 2D wall.

Material	Initial Stiffness (kN/mm)		Peak Strength (kN)		Energy Dissipation (kN.mm )	
	Value	Error (%)	Value	Error (%)	Value	Error (%)
Test	57.74	-	1561	-	14,766	-
KCC	63.23	8.68	1640	4.82	25,321	41.69
CDP	73.5	21.44	1410	−9.67	14,756	0.06
Winfrith	29.83	−48.33	1720	9.24	20,532	28.08

**Table 3 materials-17-01877-t003:** Hysteresis diagram parameters pertaining to the 3D wall for X direction.

Material	Initial Stiffness (kN/mm)		Peak Strength (kN)		Energy Dissipation (kN.mm )	
	Value	Error (%)	Value	Error (%)	Value	Error (%)
Test	49.69	-	459.01	-	85,809	-
KCC	36.98	−25.58	392.26	−14.54	66,353	−22.67
CDP	51.25	3.04	469.84	2.31	49,437	−42.44
Winfrith	37.74	−24.05	443.97	−3.28	92,114	6.84

**Table 4 materials-17-01877-t004:** Hysteresis diagram parameters pertaining to the 3D wall for Y direction.

Material	Initial Stiffness (kN/mm)		Peak Strength (kN)		Energy Dissipation (kN.mm )	
	Value	Error (%)	Value	Error (%)	Value	Error (%)
Test	29.66	-	452.06	-	106,027	-
KCC	27.05	−8.8	386.01	−14.61	101,223	−4.53
CDP	27.11	−8.6	434.87	−3.81	98,357	−7.23
Winfrith	21.69	−26.87	475.95	5.02	127,165	6.62

**Table 5 materials-17-01877-t005:** The comparison of the improvement of lateral load between the various concrete models for the 2D wall.

Concrete Strength (MPa)	Increasing Strength (%)	KCC Model		CDP Model		Winfrith Model	
		Load (kN)	Improvement (%)	Load (kN)	Improvement (%)	Load (kN)	Improvement (%)
43.4	-	1545	-	1589	-	1642	-
54.3	25	1612	4.3	1634	2.8	1724	5.1
65.1	50	1680	8.7	1750	10.1	1812	10.4
86.8	100	1797	16.3	1891	19.1	1952	18.9

**Table 6 materials-17-01877-t006:** The comparison of the improvement of lateral load between the various concrete models for the 3D wall in X-direction.

Concrete Strength (MPa)	Increasing Strength (%)	KCC Model		CDP Model		Winfrith Model	
		Load (kN)	Improvement (%)	Load (kN)	Improvement (%)	Load (kN)	Improvement (%)
45	-	368	-	455	-	446	-
56.3	25	379	3.1	471	3.5	461	3.3
67.5	50	395	7.3	491	7.9	482	8.1
90	100	410	11.4	517	13.6	503	12.8

**Table 7 materials-17-01877-t007:** The comparison of the improvement of lateral load between the various concrete models for the 3D wall in Y-direction.

Concrete Strength (MPa)	Increasing Strength (%)	KCC Model		CDP Model		Winfrith Model	
		Load (kN)	Improvement (%)	Load (kN)	Improvement (%)	Load (kN)	Improvement (%)
45	-	463	-	436	-	417	-
56.3	25	478	3.2	449	3	431	3.4
67.5	50	494	6.7	470	7.8	451	8.1
90	100	512	10.6	522	19.7	508	21.8

## Data Availability

Data are contained within the article.

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
