# Peer review of "Effective Prediction of Concrete Constitutive Models for Reinforced Concrete Shear Walls under Cyclic Loading"

_materials, 2024, doi:10.3390/ma17081877_

Round 1

Reviewer 1 Report

Comments and Suggestions for Authors

In general, the article is of scientific and practical interest and may be published in the journal Materials after correction. The authors need to significantly shorten the article, primarily in terms of describing informational material, because everyone already knows this. In addition, it is necessary to adjust the structure of the article, and add statistical evaluation and mathematical equations.

1. In the abstract, it is necessary to indicate the results of the research in numerical form.

2. Unclear numbering of sections, 0.Introduction; 0.1. Background; 0.2. Research purpose?

3. In my opinion, it is necessary to remove section 1. Constitutive modeling of concrete, because it is informative.

4. The article should include a statistical analysis of the obtained results, as well as mathematical equations.

5. In the Conclusions section, it is necessary to add several suggestions regarding the further research plan.

6. It is necessary to add several references to modern scientific works (2023-2024) to the list of literature.

Author Response

The authors would like to thank the reviewers for their positive comments and feedback regarding this manuscript on the Effective Prediction of Concrete Constitutive Models for Reinforced Concrete Shear Walls under Cyclic Loading 

Reviewer 2 Report

Comments and Suggestions for Authors

Dear authors,

His manuscript is understandable. It contains an excess of Figures and Tables, however I think they are relevant.

In order to polish your manuscript I suggest the following:

1. Avoid referencing in bulk: see lines in 53 and 59 for example.

2. In the text put the word Figure instead of Fig.

3. Expand the information in detail what the authors of the reference [44] indicate, in addition to indicating where the value 4.444Gf comes from.

4. Which are the four parameters established by Ottosen? Explain in the text of the manuscript.

5. Which is equation 2.1-7 of the CEB-FIP model code? Explain it in the text.

6. Improve the explanation 372 to 375 so that the reader does not get lost (for example; Figure 15 together with Table...

7. Table 3, Table 4, Table 6, and Table 7 are not detailed in the text.

Author Response

(The authors gave the same response as above.)

Reviewer 3 Report

Comments and Suggestions for Authors

Interesting work and well prepared. Since I'm a person from outside the industry, some of my comments may be a little strange, but that's why I can be unbiased.

You use the terms 2D and 3D walls. For me, 2D, as for most people, means something two-dimensional with an X and Y axis without height, while 3D is a spatial structure.

  As a reviewer, I focused on such ambiguities. But congratulations on a good job.

Abstract

And what results did you obtain during the conducted simulations? You will enter numerical values. The abstract is the most important part of the manuscript, if someone reads it, they start with the abstract, if they are not interested, they will not read your manuscript and quote it.

  Drauga draws conclusions from research and practical conclusions. Nowadays, every research should have a practical conclusion, e.g. after analyzing specific materials, it results that the best properties have......, which allows this material to be used in the construction of, for example, spaceships, cars, and buildings. ....

Introduction

Line 19

The material known as concrete

You write for magazines describing various types of materials from various fields, so it is worth explaining the basic concepts at the beginning whether concentrate refers to building materials or is it, for example, a mixture of dyes in a polymer or something else.

https://en.wikipedia.org/wiki/Concrete

line 26

Reinforced concrete (RC) – again it will be good to explain what kind of reifoced is used steel, glass fibers, resin fibers?

 Line 30

ACI 318 specification- please add direct ref to the standard or write ACI 318 specification describe in the article

Line 52

two-dimensional finite element (FE)

three-dimensional finite element (FE) – you are using the same aberration for two different  definition, please correct it.

Equation 1-3

What the a values mean?

Line 160

eight longitudinal bars made with steel?

What were the limitations of your research? What else would you like to test next? Practical conclusion?

Good luck in your further research!

References

Seible, F.; LaRovere, H.L.; Kingsley, G.R. Nonlinear analysis of reinforced concrete masonry shear wall structures—monotonic loading. The Masonry Society Journal 1990, 9. Missing pages,

Check all refs again so that first there is the name of the journal, year, journal number and page number,

Hong, J.; Fang, Q.; Chen, L.; Kong, X. Numerical predictions of concrete slabs under contact explosion by modified K&C material  model. Construction and Building Materials 2017, 155, 1013–1024.  

37. Xu, M.; Wille, K. Calibration of K&C concrete model for UHPC in LS-DYNA. In Proceedings of the Advanced materials research. 553 Trans Tech Publ, 2015, Vol. 1081, pp. 254–259- use the same citation style, please

Author Response

(The authors gave the same response as above.)

Round 2

Reviewer 1 Report

Comments and Suggestions for Authors

I think that you have done a great work.